# The Selection of NFκB Inhibitors to Block Inflammation and Induce Sensitisation to FasL-Induced Apoptosis in HNSCC Cell Lines Is Critical for Their Use as a Prospective Cancer Therapy

**DOI:** 10.3390/ijms20061306

**Published:** 2019-03-15

**Authors:** Mario Joachim Johannes Scheurer, Roman Camillus Brands, Mohamed El-Mesery, Stefan Hartmann, Urs Dietmar Achim Müller-Richter, Alexander Christian Kübler, Axel Seher

**Affiliations:** 1Department of Oral and Maxillofacial Plastic Surgery, University Hospital Würzburg, D-97070 Würzburg, Germany; marioscheurer@icloud.com (M.J.J.S.); Brands_R@ukw.de (R.C.B.); Hartmann_S2@ukw.de (S.H.); Mueller_U2@ukw.de (U.D.A.M.-R.); Kuebler_A@ukw.de (A.C.K.); 2Comprehensive Cancer Center Mainfranken, University Hospital Würzburg, D-97080 Würzburg, Germany; 3Department of Biochemistry, Faculty of Pharmacy, Mansoura University, Mansoura E-35516, Egypt; elmesery@hotmail.com; 4Interdisciplinary Center for Clinical Research, University Hospital Würzburg, D-97070 Würzburg, Germany

**Keywords:** HNSCC, NFκB, inhibitor, TPCA1, apoptosis, inflammation, TNFα, FasL

## Abstract

Inflammation is a central aspect of tumour biology and can contribute significantly to both the origination and progression of tumours. The NFκB pathway is one of the most important signal transduction pathways in inflammation and is, therefore, an excellent target for cancer therapy. In this work, we examined the influence of four NFκB inhibitors—Cortisol, MLN4924, QNZ and TPCA1—on proliferation, inflammation and sensitisation to apoptosis mediated by the death ligand FasL in the HNSCC cell lines PCI1, PCI9, PCI13, PCI52 and SCC25 and in the human dermal keratinocyte cell line HaCaT. We found that the selection of the inhibitor is critical to ensure that cells do not respond by inducing counteracting activities in the context of cancer therapy, e.g., the extreme IL-8 induction mediated by MLN4924 or FasL resistance mediated by Cortisol. However, TPCA1 was qualified by this in vitro study as an excellent therapeutic mediator in HNSCC by four positive qualities: (1) proliferation was inhibited at low μM-range concentrations; (2) TNFα-induced IL-8 secretion was blocked; (3) HNSCC cells were sensitized to TNFα-induced cell death; and (4) FasL-mediated apoptosis was not disrupted.

## 1. Introduction

Despite numerous innovations in cancer therapy, the successful treatment of tumours is still a major medical challenge. Over the past few decades, it has become increasingly apparent that there is no gold standard or single approach to therapy; every patient and every form of cancer must be treated specifically. However, “central hallmarks” intrinsically characterize cancer, including (a) autonomous proliferation, (b) tumour suppressor elimination, (c) migration/invasion activation and subsequent metastasis, (d) angiogenesis, (e) replicative immortality, and (f) enhanced or complete resistance to cell death [1]. Although the fate of cancer is closely linked to genes and their alleles, physiological circumstances can contribute considerably. A key role is played by the immune system and/or inflammation, which itself can be the cause of the disease and can profoundly influence all of the previously mentioned central hallmarks of cancer [2]. Thus, via treatment with anti-inflammatory agents, both the incidence and mortality of different tumour types could be reduced [3,4]. Numerous signal transduction pathways, e.g., RAS-RAF and MYC [5,6], are involved in pro-inflammatory and tumour-promoting processes, however, the nuclear factor kappa light-chain enhancer of activated B-cells (NFκB) pathway has a preeminent position in the context of cancer and inflammation.

The NFκB family of transcription factors consists of five members: NFκB1 (p50/p105), NFκB2 (p52/p100), c-Rel, RelA (p65) and RelB. The highly conserved Rel homology domain is common to all and is responsible for DNA binding as well as for protein–protein interactions. In mammalian cells, two primary signal transduction pathways play a pivotal role: the classical or canonical pathway and the alternative or noncanonical pathway. The classical pathway is induced by cytokines such as tumour necrosis factor (TNF)α or interleukin (IL)-1β, leading to the activation of the cytoplasmic classical inhibitory κB (IκB) kinase complex (IKK), which comprises three subunits (IKK-α/β/γ). In the inactive form, NFκB heterodimers are associated with IκB. In the classical pathway, IκBα is typically associated with the p65/p50 heterodimer. The phosphorylation of IκBα by IKK leads to ubiquitination followed by proteasomal degradation and thus to the release of p65/p50, which translocates into the cell nucleus and regulates gene activity.

In contrast, the alternative pathway is mainly regulated by members of the TNF family, such as lymphotoxin-β (LTβ), receptor activator of NFκB ligand (RANKL), B-cell activating factor (BAFF) or CD40. In this pathway, ligand-mediated receptor activation leads primarily to the stabilisation of NFκB-inducing kinase (NIK). NIK selectively activates IKKα, leading to phosphorylation of the NFκB dimer p100/RelB. Proteolytic cleavage of p100 to p52 then allows the heterodimeric transcription factor p52/RelB to translocate into the nucleus [7]. The classical and alternative pathways activate different gene subsets. The classical pathway regulates the expression of genes encoding proteins responsible for the innate immune response, such as cytokines (IL-1, IL-6, IL-8 and TNFα), chemokines (monocyte chemoattractant protein 1 (MCP-1), IL-18 and CC-chemokine ligand 5 (CCL-5)), anti-apoptotic factors (cellular FLICE-like inhibitory protein (cFLIP), inhibitor of apoptosis proteins (IAPs) and Survivin), cell cycle regulators (cyclins) and adhesion molecules (ICAM-1, VCAM-1 and matrix metalloproteinases (MMPs)) [8]. The alternative pathway is responsible for the regulation of adaptive immune response genes and is involved in B-cell development and maturation, T-cell differentiation via the induction of cytokines such as IL-12, and the development of secondary lymphoid organs [9,10,11].

These abovementioned genes play important roles in the development of cancer. Thus, NFκB pathway inhibition is an excellent approach for cancer therapy. The relevance of this pathway in tumour biology has been shown in different mouse models; for example, the deletion of IKK-β in enterocytes decreased the tumour incidence in a colitis-associated colon cancer mouse model [12], and overexpression of IκBα in liver cells in a hepatocarcinoma model blocked tumour development [13]. Moreover, the NFκB pathway was required for the development of tumours in a mouse model of lung adenocarcinoma, and loss of p53 and expression of oncogenic KRAS resulted in NFκB activation [14]. The relevance of NFκB has also been shown in humans. Bortezomib (PS341), a dipeptide boronic acid, inhibits the proteasome complex. Inhibition of NFκB through the accumulation of IκBs is thought to play a key role in this process. Indeed, bortezomib is already approved for the treatment of multiple myeloma [15,16]. Another example of the relevance of NFκB in human cancer is IKK inhibition by aspirin. Aspirin blocked the initiation and progression of colorectal cancer [3,17].

Another key point is the known involvement of the NFκB pathway in resistance mechanisms of cancer cells. On one hand, by the previously described intrinsic mechanisms, NFκB pathway activation promotes cancer by creating an inflammatory milieu and inducing anti-apoptotic genes [18]. On the other hand, extrinsic activation or promotion of the NFκB pathway by chemotherapeutic agents can facilitate resistance and thus counteract the therapy [19,20]. This phenomenon is thoroughly demonstrated for the cytostatic agent cisplatin. NFκB activity is inversely correlating with cisplatin sensitivity in cancer cell lines. In addition, prolonged treatment with cisplatin may lead to cisplatin resistance, which is correlated with increased activation of NFκB [21].

In head and neck squamous cell carcinoma (HNSCC), the NFκB pathway also plays an important role through strong or constitutive pathway activation [22,23], which is accompanied by high expression/release of inflammatory factors [23,24,25,26] and is partly responsible for tumour progression [27]. HNSCC is one of the most common tumour entities, with over 600,000 new cases diagnosed annually worldwide. The poor prognosis, with a 5-year survival rate of approximately 55%, is striking [28]. In recent decades, the prognosis has not improved, despite intensive research and the introduction of new adjuvant therapies [28,29]. Thus, developing new strategies for treating HNSCC is imperative, and, for the abovementioned reasons, the NFκB pathway is an ideal target. In addition, a variety of potential agents that intervene at various points in NFκB signal transduction have been developed. The IKK complex in the classical NFκB pathway or NIK in the alternative pathway are frequent targets [30].

In this work, we examined the influence of the four NFκB inhibitors Cortisol, MLN4924, QNZ and TPCA1 on proliferation, inflammation and sensitisation to apoptosis mediated by the death ligand FasL in the HNSCC cell lines PCI1, PCI9, PCI13, PCI52 and SCC25 [18] and in the human keratinocyte cell line HaCaT.

## 2. Results

Since NFκB-mediated inflammation, as described above, is a major factor promoting tumour formation as well as contributing to increased apoptosis resistance in tumours, the effect of four mechanistically different NFκB inhibitors—Cortisol, MLN4924, QNZ and TPCA1—on the secretion of inflammatory cytokines (IL-8 and TNFα), proliferation (relative cell number (RCN)) and sensitisation to death ligand (FasL)-induced apoptosis was analysed in the HNSCC cell lines PCI1, PCI9, PCI13, PCI52 and SCC25 as well as in the keratinocyte cell line HaCaT as the control/standard.

Cortisol belongs to the glucocorticoids and is often used to counteract inflammatory and sometimes chronic processes, e.g., rheumatism or gout. Although its exact mechanism of action is unknown, Cortisol leads to the inhibition of IκBα degradation and NFκB p65 phosphorylation and thus to the suppression of NFκB target genes such as cyclooxygenase-2 (COX-2), TNFα, IL-1β, and IL-6 [31]. MLN4924 represses the ubiquitin proteasome system by inhibiting neural precursor cell expressed, developmentally downregulated 8 (NEDD8) activating enzyme (NAE) [32]. This molecule acts as a potent inhibitor of NFκB activation by, e.g., inhibiting the degradation of IκBα, and the reduced p100 processing leads to NIK accumulation [33,34]. MLN4924 suppresses the growth of and radio sensitizes HNSCC cells and tumours [35]. QNZ shows strong inhibitory effects on both NFκB transcriptional activation and TNFα production, although the mechanism is unknown. Furthermore, QNZ exhibited an anti-inflammatory effect on carrageenan-induced paw oedema in rats [36]. TPCA1 is mainly an IKKβ inhibitor and, therefore, potently influences the classical pathway by inhibiting the phosphorylation and degradation of IκBα by inhibiting the expression of NFκB target genes such as COX-2, IL-6 and IL-8 [33,37].

### 2.1. HNSCC Cell Lines Secrete the Pro-Inflammatory Marker IL-8 and Are Responsive to TNFα

In HNSCC, the NFκB pathway is important because of its strong constitutive activation [22,23] accompanied by robust initiation of inflammatory factor production [23,24,25,26], thus promoting tumour progression [27]. In the first experiment, we analysed the pro-inflammatory status in our cell lines by measuring the secretion of IL-8, a well-established marker of inflammation and NFκB pathway activity. IL-8 secretion was found in the supernatant of all analysed cell lines after 24 and 48 h. Additionally, all HNSCC cell lines were responsive to TNFα and exhibited a significant increase in IL-8 secretion after 24 and 48 h (Figure 1A). To exclude the possibility that TNFα-induced cell proliferation/cell death was responsible for the difference in IL-8 secretion, the relative cell number was determined via crystal violet staining. TNFα induced no or only a slight increase in cell proliferation (Figure 1B).

### 2.2. Inhibition of HNSCC Cell Proliferation by NFκB Inhibitors

In the next experiment, the effect of the four NFκB inhibitors on proliferation was investigated. Figure 2 summarises the results for each individual inhibitor. The determined IC_10_ (10% inhibition with respect to the relative cell number) and IC_50_ (50% of the maximum inhibition) values are summarised in Table 1. Cortisol showed the weakest antiproliferative effect in all tested cell lines. At the highest concentration (50 μg/mL), the relative cell number was only reduced to values between 62% and 91% of baseline (Table 2). In terms of the IC_50_, the next most potent inhibitor was TPCA1, with IC_50_ values between 5.3 and 24.2 μM. Notably, all cell lines were responsive to TPCA1, and the relative cell numbers were reduced to values between 10.6% and 29.9% of baseline. QNZ was more effective than Cortisol and TPCA1, with IC_50_ values between 1 and >10 μM, but only 3 of the 6 cell lines (PCI1, PCI13, and HaCaT) responded with a reduction in the relative cell number of 50% or greater. In those cell lines, the cell number was reduced to values between 5.7% and 96.3%. MLN4924 exhibited the lowest IC_50_ values (between 0.8 and 17.4 μM); the relative cell number was reduced to values between 9.9% and 39.3%.

### 2.3. Inhibition of the Inflammatory Response in HNSCC Cells by NFκB Inhibitors

The aim of the next experiment was to analyse the potential of NFκB inhibitors to block inflammation. Cells were incubated with the indicated concentrations of the respective inhibitors with and without TNFα (5 ng/mL) for 48 h, and the amount of secreted IL-8 in the supernatant was subsequently determined by ELISA. IL-8 is an established marker of both inflammation and NFκB pathway activity [26,38]. The TNFα concentration of 5 ng/mL was chosen from earlier evaluation experiments showing that this relatively low TNFα concentration is sufficient to stimulate IL-8 production without overstimulating the NFκB pathway and thus abrogating the efficacy of the inhibitors. The results are summarised in Figure 3. Cortisol had no significant effect on either basal or TNFα-induced IL-8 secretion, and QNZ exhibited a similar pattern. While the basal level was only slightly increased, TNFα-induced IL-8 secretion could not be inhibited. However, the results for MLN4924 were pronounced. Although TNFα-induced IL-8 secretion was not significantly inhibited, treatment with MLN4924 alone led to a massive increase in the basal IL-8 level in all HNSCC cell lines. Compared to the other three NFκB inhibitors, TPCA1 exhibited a completely different effect. The basal IL-8 level was lowered slightly; however, crucially, TNFα-induced IL-8 secretion was blocked, remaining, at worst, at the basal level.

The HaCaT cell line should, in principle, serve as an internal standard, as it is a spontaneously immortalised keratinocyte cell line and is thus similar to the phenotype of the HNSCC cell lines in terms of the original squamous epithelium [39]. The effects of the NFκB inhibitors in this cell line were nearly negligible. Although the IL-8 level increased after incubation with the inhibitor MLN4924, the increase of 1.4-fold was significantly lower than that observed in the HNSCC cell lines.

### 2.4. TNFα Induced HNSCC Cell Death after TPCA1 Stimulation

However, in HaCaT cells, combined stimulation with TNFα and TPCA1 led to a strong reduction in the IL-8 level. In principle, TNFα can activate the classical NFκB pathway and thus influence the expression of numerous genes, both pro-apoptotic and anti-apoptotic. Inhibition of the NFκB pathway can lead to altered homeostasis of anti- and pro-apoptotic genes and render the inflammatory factor TNFα, a death ligand, which triggers apoptosis [40,41,42]. The classical experiment involved the incubation of cell lines (e.g., HaCaT) [43] with the antibiotic cycloheximide (CHX). CHX attacks ribosomes, inhibiting de novo protein synthesis and leading to cFLIP inhibition. TNFα consequently induces apoptosis [44]. The same or a similar effect can be achieved by NFκB inhibitors. For example, MLN4924 sensitizes monocytes and maturing dendritic cells to TNF-dependent and independent necroptosis, another form of programmed cell death [45].

To determine whether the combination of TNFα and TPCA1 leads to a reduction in cell number, all cell lines were stimulated with the cell line-specific IC_10_ of TPCA1 and 100 ng/mL TNFα for 72 h (Figure 4). In principle, TPCA1 reduced the relative cell number in all cell lines, thus not only blocking IL-8 secretion via the classical NFκB pathway but also allowing the induction of cell death via TNFα.

### 2.5. Analysis of Extrinsic FasL-Induced Apoptosis in Combination with NFκB Inhibitors in HNSCC Cells

In a final experiment, we investigated whether the NFκB inhibitors used in this study are suitable for sensitising HNSCC cell lines to FasL-induced extrinsic apoptosis. Cell lines were stimulated with the cell line-specific IC_10_ of the NFκB inhibitors (Table 1) and increasing concentrations of the death ligand FasL. Previous studies from our group showed that the cell lines used in this study express the Fas receptor and can be sensitized to FasL-mediated apoptosis by the SMAC mimetics Birinapant and LCL161, e.g., [46,47].

Detailed results for Cortisol are shown in Figure 5. An overview of all cell lines and inhibitors is provided in Table 3, which shows an analysis of sensitisation to apoptosis as numerical values for the enhancement (*x* times) and the interaction index (*y*) described by Tallarida [48].

In summary, combination therapy with NFκB inhibitors and FasL exhibited no significant effect in most cases. However, importantly, in some cases, the combination inhibited or strongly negatively affected FasL-induced apoptosis. This effect is exemplified for Cortisol in Figure 5. The cell lines PCI9, PCI52 and SCC25, which are generally not responsive to FasL, showed no sensitisation to apoptosis in combination with Cortisol. The two FasL-responsive cell lines PCI1 and PCI13 even showed a significant deterioration in apoptotic efficiency. While the efficiency in PCI1 cells was reduced by 60%, PCI13 cells showed newly gained FasL resistance. The values in Table 3 show that most of the cell lines, regardless of the inhibitor used, showed no significant changes in FasL sensitivity. The only two exceptions were QNZ in PCI13 cells, which led to a 2.3-fold improvement in efficiency, and MLN4924 in SCC25 cells, which led to a 5-fold improvement. These results contrast with the results in the control cell line HaCaT. Basically, sensitisation to FasL-induced apoptosis by NFκB inhibitors is possible. Here, with the exception of Cortisol, all the inhibitors used imparted a clear increase in the efficiency of FasL-induced apoptosis in HaCaT cells, with increases of 2.4-fold for QNZ, 2.5-fold for TPCA1 and 3.9-fold for MLN4924.

## 3. Discussion

Inflammation is a central aspect in tumour biology and can contribute significantly to both the origin and progression of tumours. Although the importance of inflammation in the context of cancer has long been known, this aspect has become increasingly important due to NFκB inhibitors. The NFκB pathway is one of the most important signal transduction pathways in inflammation and is, therefore, an excellent target for cancer therapy.

In the last three decades, no progress has been made in HNSCC treatment; thus, the prognosis is usually poor, with a five-year survival rate of 55% [28,29]. Since HNSCC is also characterized by massive inflammation and strong activation of the NFκB pathway, NFκB inhibitors seem particularly useful for this disease. By selecting Cortisol, MLN4924, QNZ and TPCA1, we adopted four inhibitors that attack different nodes within the NFκB signaling pathway. We sought to analyse the effectiveness of these inhibitors in suppressing inflammation, reducing proliferation, and sensitising to apoptosis mediated by the death ligand FasL in five HNSCC cell lines.

Of the four inhibitors used in this study, only TPCA1 truly qualifies for clinical use in HNSCC. While Cortisol had little effect in the proliferation assays, QNZ, MLN4924, and TPCA1 potently inhibited cell proliferation, with IC_50_ values in the lower μM range. Although these concentrations are still quite high for the use of these agents as purely antiproliferative drugs, the antiproliferative effect can be considered as an additional positive effect. With regard to use as anti-inflammatory therapeutic agents, neither Cortisol nor QNZ showed a significant effect on decreasing the IL-8 level. Even more strikingly, the effect of MLN4924 on the HNSCC cells was counterproductive due to the strong increase in IL-8 secretion. Thus, because of the aim to use NFκB inhibitors as a therapeutic approach to reduce inflammation, MLN4924 has been disqualified for use in HNSCC treatment based on our in vitro data. TPCA1, however, showed inhibition of TNFα-induced IL-8 secretion. TNFα is often an essential tumour promoter through mediating the production of pro-inflammatory factors.

A quite surprising result was found in the attempt to enable FasL-mediated apoptosis via NFκB inhibitors. Basically, the inhibitors showed almost no beneficial effects, except for QNZ, which led to a 2.3-fold improvement in efficiency in PCI13 cells, and MLN4924, which led to a 5-fold improvement in SCC25 cells. The results for Cortisol are extremely important due to the observed increase in apoptosis resistance, even in the two FasL-responsive cell lines, PCI1 and PCI13, which showed a significant deterioration in apoptotic efficiency. The efficiency in PCI1 cells was reduced by 60%, and PCI13 cells showed newly gained FasL resistance.

The finding that TPCA1 can facilitate TNFα-mediated apoptosis is completely new. Importantly, this effect was observed in all of the studied cell lines. In HNSCC, TNFα can inhibit apoptosis by activating the Akt serine/threonine kinase [49]. However, altered homeostasis of intracellular factors may render TNFα pro-apoptotic. This dual Janus-like role of TNFα has long been known [50]; examples are the CHX-induced degradation of the anti-apoptotic factor cFLIP [42] and the use of NFκB inhibitors in multiple myeloma cells [45], which enables the activation of programmed cell death by TNFα.

In vivo studies in HNSCC have shown that nonsteroidal anti-inflammatory drugs are beneficial for patient survival [51]. Among the four inhibitors—Cortisol, MLN4924, QNZ and TPCA1—only a few data directly related to HNSCC have been published. Thus, Cortisol has no significant advantage for use in HNSCC therapy [52,53]. MLN4924 has already been shown to increase apoptosis in HNSCC, as demonstrated by the decrease in the level of cFLIP, which promotes TRAIL-induced apoptosis in HNSCC cells [54]. Another group showed that MLN4924 induced significant replication and inhibited HNSCC cell proliferation both in culture and in HNSCC xenografts in mice. Additionally, MLN4924 sensitizes HNSCC cells to ionising radiation (IR) and enhances IR-induced suppression of xenograft growth in mice [35]. Zhang et al. provided more detailed insight into the mechanism of MLN4924-induced apoptosis, as follows. The expression levels of NAE1 and UBC12 were higher in HNSCC tissues than in normal tissues, and inactivation of the neddylation pathway significantly inhibited malignant phenotypes in HNSCC cells. Mechanistic studies revealed that MLN4924 induced the accumulation of the CRL ligase substrate c-Myc, which transcriptionally activated the pro-apoptotic protein Noxa and triggered apoptosis in HNSCC [55]. The finding that MLN4924 induces massive IL-8 secretion in HNSCC cells is new. The mechanism is unclear, but the use of this agent in cancer therapy must be tested intensively, since IL-8 secretion will be extremely counterproductive—as shown in HNSCC, IL-8 may contribute to tumour progression [26,56]. In an additional experiment, we tested the hypothesis that MLN4924 can induce the secretion of soluble TNFα and by this way induce the autocrine/paracrine secretion of IL-8. The analysis of the HNSCC cell lines after 24 and 48 h of stimulation with MLN4924 showed no significant detectable TNFα signal (except positive standard control) in stimulated and unstimulated probes (data not shown). We found no data in the literature regarding QNZ and TPCA1 in the context of HNSCC.

How should the results of this in vitro study be evaluated in terms of their basic informative value? We believe that the use of Cortisol and MLN4924 on the basis of the in vitro data is critical. Although Cortisol showed no relevant effect in most cases, the increase in apoptosis resistance in 2 of the 5 tested HNSCC cell lines must be considered. Additionally, the increase in IL-8 secretion induced by MLN4924 must be assessed very carefully. Especially compared with the control cell line HaCaT, the HNSCC cell lines exhibited clearly different responses to the inhibitors. In HaCaT cells, NFκB inhibitors resulted in a minimal increase in IL-8 secretion (a maximal increase of 1.4-fold). Moreover, regarding FasL-induced apoptosis MLN4924, QNZ and TPCA1 resulted in a significant increase in FasL responsiveness in HaCaT cells; thus, the NFκB inhibitors showed the precise expected effects.

Why Cortisol has such a drastic counteracting effect in HNSCC cannot be clearly explained. As already mentioned, two studies have demonstrated the inefficacy of Cortisol in HNSCC for the induction of apoptosis [53] as well as the inhibition of proliferation [52]. For MLN4924, we could not find comparable results in the literature for the observed induction of IL-8 secretion. Certainly, further studies are needed to elucidate the molecular mechanisms underlying these results to rule out the use of Cortisol and MLN4924 or to limit their use only for specific patients. QNZ effectively inhibited proliferation at low concentrations, but this finding was the only convincing result in favour of this inhibitor. TPCA1, however, qualified as a suitable NFκB inhibitor in HNSCC in all experiments and all cell lines by four positive qualities: (1) Proliferation was inhibited by low μM range concentrations of TPCA1; (2) TNFα-induced IL-8 secretion was blocked by TPCA1; (3) HNSCC cells were sensitised to TNFα-induced cell death by TPCA1; and (4) TPCA1 did not interfere with FasL-mediated apoptosis.

But why is TPCA1 so effective compared to the other NFκB inhibitors? Based on the current state of knowledge, the effect can’t be explained by a detailed molecular mechanism. However, TPCA1 attacks the IKK2 complex specifically, while the other inhibitors affect only indirectly the IκBα phosphorylation and/or the NFκB pathway. We assume that in the analysed HNSCC cells TPCA1 is the only inhibitor that efficiently inhibits the classical NFκB pathway—TPCA1 alone was able to inhibit IL-8 secretion. The other drugs may affect the classical or alternative pathway, but not specifically. In HNSCC cells, they are more responsible for an imbalance in NFκB-regulated gene expression and a rather unspecific cellular response. Furthermore, the influence of other signal transduction pathways is also important. Cortisol, in its active form, is a transcription factor that can activate many gene products, the exact mechanism of QNZ is unknown, and MLN4924 also does not specifically interfere via the proteasomal degradation. Thus, the main advantage of TPCA1 is its selectivity, which also contributes to the fact that this drug has no effect on FasL-mediated apoptosis in HNSCC cell lines.

We propose that the in vitro data collected in this publication indicate that TPCA1 is suitable for clinical use. Further in vivo studies are needed to demonstrate the efficacy of this inhibitor in HNSCC.

## 4. Materials and Methods

All inhibitors of the NFκB signaling pathway analysed in this study were commercially purchased. MLN4924 (pevonedistat; Hycultec, Beutelsbach, Germany), QNZ (EVP4593) and TPCA1 (both Absource Diagnostics, Munich, Germany) were dissolved in dimethyl sulfoxide (DMSO; Sigma-Aldrich, Darmstadt, Germany) to 10 mM stock solutions. Cortisol was obtained as a ready-to-use solution in methanol (1 mg/mL) (Sigma-Aldrich, Darmstadt, Germany). Stock solutions were stored at −20 °C. In preliminary experiments, the toxicity of DMSO and methanol was determined. The maximum concentrations used in the experiments for DMSO (<=1%) and for methanol (<=5%) showed no significant toxic effects (reduction of the relative cell number) in the cell lines we used. Ligand preparation of Fc-FLAG-FasL and wild-type TNFα-FLAG was performed as previously described [57].

### 4.1. Cell Culture

The HNSCC cell lines PCI1, PCI9, PCI13 and PCI52 were established at the Cancer Institute of the University of Pittsburgh (USA) [58] and have already been used in several studies, particularly in studies of the cytotoxicity of antineoplastic drugs [47,58,59,60]. Cell lines were cultured in Dulbecco’s modified Eagle’s medium with low glucose (DMEM low; Thermo Fisher, Karlsruhe, Germany) supplemented with 10% fetal calf serum (FCS; Thermo Fisher), 1% penicillin/streptomycin (P/S; 100 U/mL penicillin and 100 µg/mL streptomycin; Thermo Fisher) and 5 mL glutamine (4 mM; Biochrom, Berlin, Germany). SCC25 and HaCaT cells were purchased from the American Type Culture Collection (ATCC). SCC25 cells were cultured in DMEM/F-12 supplemented with HEPES (Thermo Fisher), 10% FCS, 1% P/S and 5 µL of hydrocortisone (400 ng/mL; Sigma-Aldrich). HaCaT cells were cultured in DMEM supplemented with 10% FCS and 1% P/S (Thermo Fisher). Cells were cultured at 37 °C in a humidified atmosphere containing 5% CO_2_.

### 4.2. Crystal Violet Staining (CytoTox) Assay

Cells were seeded in triplicate in 100 µL of culture medium (1 × 10^4^ cells/well) overnight. The following day, cells were incubated with NFκB inhibitors at the indicated concentrations for 72 h. For staining, supernatants were removed, and cells were incubated with 50 µL/well crystal violet solution (1% crystal violet in 20% methanol; Carl Roth, Karlsruhe, Germany) for 12 min and subsequently washed five times with distilled water. The plates were dried for 24 h in the dark. For quantification, 100 µL/well methanol was added and incubated for a 12 min until the crystal violet was completely dissolved. Photometric absorbance was measured at 595 nm using a microplate reader (Tecan, Crailsheim, Germany). For data analysis the experiments were repeated three times to calculate mean values and standard deviations (*n* = 3). The results were normalised to the untreated control (100%). Therefore, the relative cell number values determined via the crystal violet assay with the stimulated probes (CV**_S_**) were normalised to those of the untreated control (CV**_C_**) ((CV**_S_**/CV**_C_**) = CV**_R_**). To obtain percentage values, CV**_R_** was multiplied by 100 (RCN (%) = (CV**_S_**/CV**_C_**) × 100 = CV**_R%_**). For statistical analysis, these results were evaluated with Student’s *t*-test in order to show significant sensitising effects of the investigated NFκB inhibitors in combinational treatment with FasL. The significance level was set to *p* < 0.05.

### 4.3. ELISA

The anti-inflammatory effect of the NFκB pathway inhibitors was tested by ELISA under unstimulated or TNFα-stimulated conditions or under treatment with NFκB inhibitors alone at the indicated concentrations. A total of 2 × 10^4^ cells/well were seeded in 100 µL/well culture medium in 96-well plates. The following day, the medium was removed, and the cells were pre-incubated in medium containing Cortisol, MLN4924, QNZ or TPCA1 for 6 h. After pre-incubation, TNFα was added. Secretion of IL-8 and TNFα was investigated with ligand-specific ELISA kits (BD Biosciences, Heidelberg, Germany). After 24 and 48 h, supernatants were harvested from duplicate wells and stored at −20 °C until ELISA was performed. ELISA was carried out in accordance with the manufacturer’s protocol. Briefly, capture antibodies were coated on a clear, high-binding ELISA plate (Greiner Bio-One, Frickenhausen, Germany) overnight at 4 °C. Supernatants were incubated on the plate for 2 h and, after a wash procedure, treated with streptavidin-horseradish peroxidase-conjugated detection antibodies. After five intensive washes, 100 µL/well enzyme substrate (a 2:1 mixture of solutions A and B of the “substrate pack” (BD Biosciences)) was added, and the reaction was stopped with stop solution (2 N H_2_SO_4_) after 10 to 30 min. The results were quantified at 405 nm with an ELISA plate reader (Tecan) and normalised to the ligand standard curve in ng/mL by linear regression analysis. For standardisation, cytokine concentrations were correlated with the relative cell number quantified by the crystal violet assay. Therefore, the concentration value for each secreted cytokine determined via ELISA with the stimulated probes (ELISA**_S_**) was normalised to the corresponding value determined in the control ELISA (ELISA**_C_**) to obtain the relative expression level (ELISA**_S_**/ELISA**_C_** = ELISA**_R_**). To normalise the cytokine expression values to the relative cell numbers, the quotient ELISA**_R_**/CV**_R_** = E**_R_** was determined for each value. In this case, the results for the ELISA and the crystal violet assay were determined from the same 96-well plate. For statistical analysis, three independently performed experiments were reproduced (*n* = 3) and statistics were calculated by Wilcoxon rank-sum test. *p* < 0.05 indicates statistically significant effects marked with *, highly significant *p*-values (<0.01) with ** and very highly significant *p*-values (<0.001) marked with ***.

### 4.4. Statistical Analysis

Data collection and plotting were performed with Excel (Microsoft, Redmond, WA, USA) and GraphPad Prism (version 6.04; GraphPad Software, San Diego, CA, USA) software. Statistical analysis was accomplished with GraphPad Prism and MEDAS (Grund EDV-Systeme, Margetshöchheim, Germany) software. For analysis of the NFκB pathway inhibitor sensitisation effects, the half maximal inhibitory concentration (IC_50_) was determined by for each drug and cell line using GraphPad Prism. The IC_50_ is defined as the concentration that reach half maximal activity. For the combination treatments, the initial inhibitory concentration (IC_10_) was determined as the concentration that reduced the relative cell count by 10%. If no IC_10_ could be determined, the lowest concentration with significant cytotoxicity was defined as the IC_10_ value. Via Student’s *t*-test, we compared the effects of FasL monotherapy with those of combination treatment with FasL and NFκB pathway inhibitors. Here, *p*-values < 0.05 indicated statistically significant effects of the combination therapy. The quantitative assessment was evaluated by the interaction index (*y*) described by Tallarida [36] to identify synergistic, additive or antagonistic effects. ELISA results were analysed using the nonparametric Wilcoxon matched pairs signed rank test. A significance level of *p* < 0.05 was established to indicate statistically significant effects. ELISA results were analysed using the Wilcoxon rank-sum test. A significance level of *p* < 0.05 was established to indicate statistically significant effects and marked with *, highly significant *p*-values (<0.01) with ** and very highly significant *p*-values (<0.001) marked with ***.

## Figures and Tables

**Figure 1 ijms-20-01306-f001:**
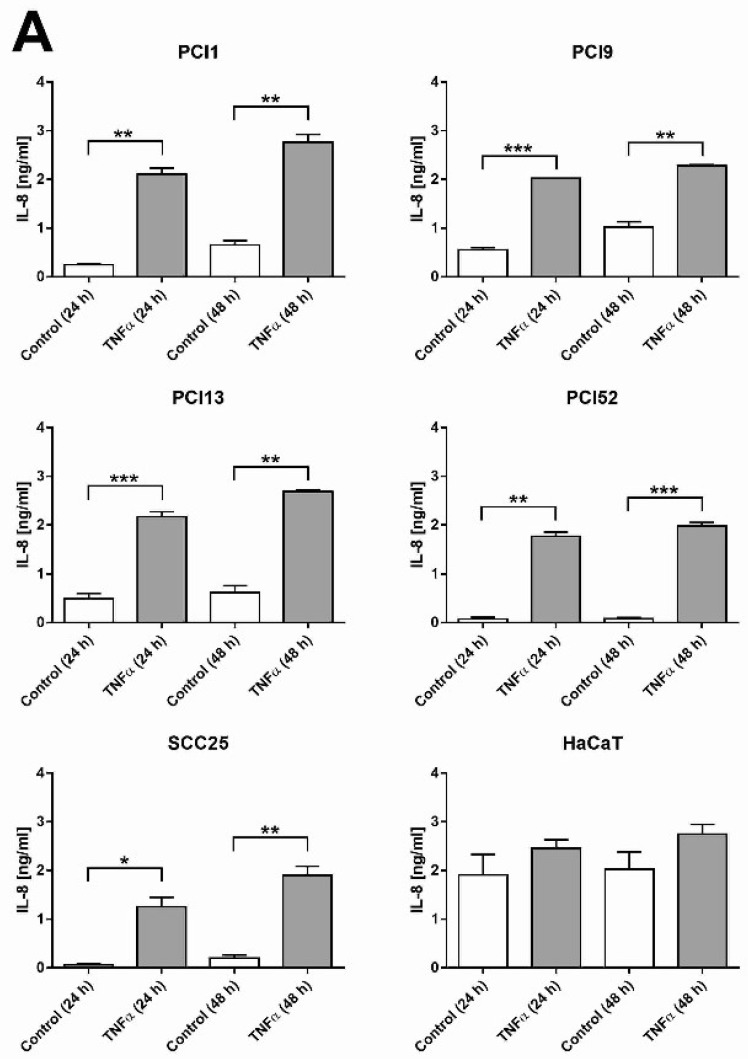
Cells (2 × 10^4^/well) were stimulated with or without TNFα (100 ng/mL) for 24 and 48 h. (**A**) IL-8 secretion was measured via ELISA of the supernatants. (**B**) Relative cell number (RCN) was measured via a crystal violet assay of the corresponding 96-well plate. RCN values were normalized to those of the 24 h control group (100%). Three independent experiments were performed to determine mean values and standard deviations (*n* = 3), which are presented above. Results were calculated by Wilcoxon rank-sum test. *p* < 0.05 indicates statistically significant IL-8 inducing effects of TNFα marked with *, highly significant *p*-values (<0.01) with ** and very highly significant *p*-values (<0.001) marked with ***.

**Figure 2 ijms-20-01306-f002:**
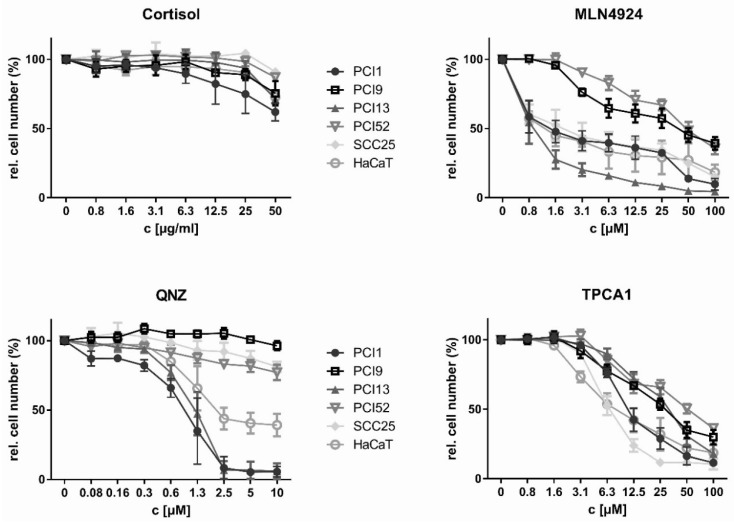
Inhibition of proliferation by Cortisol, MLN4924, QNZ and TPCA1 in HNSCC cell lines. Cells (1 × 10^4^/well) were stimulated for 72 h with the indicated concentrations of Cortisol, MLN4924, QNZ and TPCA1. The RCN was determined via crystal violet staining and normalised to that of the untreated control (100%). Mean values of the experiments reproduced at three independent time points (*n* = 3) are shown. For efficacy evaluation, the IC_50_ was determined for each NFκB inhibitor and cell line.

**Figure 3 ijms-20-01306-f003:**
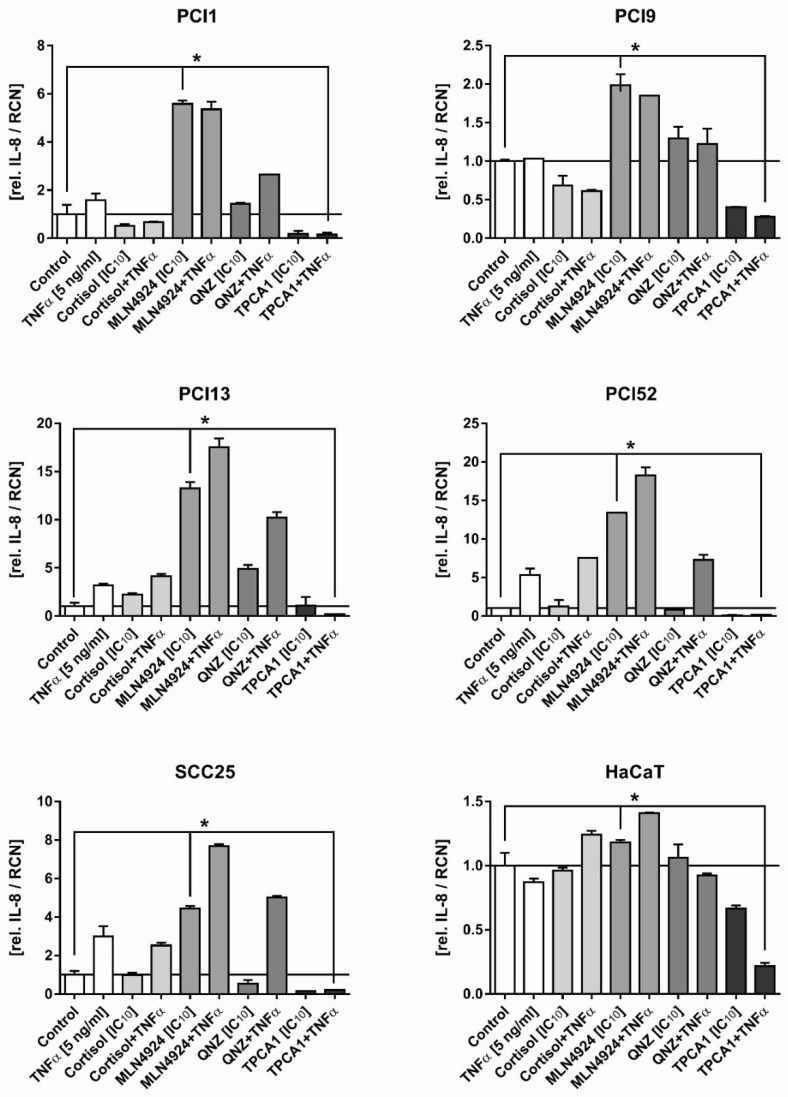
IL-8 secretion from HNSCC cell lines after incubation with the NFκB inhibitors Cortisol, MLN4924, QNZ and TPCA1 alone and in combination with TNFα. Cells (2 × 10^4^/well) were preincubated for 6 h with the appropriate IC_10_ of each inhibitor and subsequently incubated for an additional 48 h with and without TNFα (5 ng/mL). IL-8 values were normalised to RCNs determined in a crystal violet assay of the corresponding 96-well plate, as described in the material and methods section. Representative results from a total of three independently reproduced experiments (*n* = 3) are presented. The Wilcoxon rank-sum test was used for statistical data evaluation. *p* < 0.05 depicts statistically significant IL-8 inducing or inhibiting effects of the indicated treatment by taking the corresponding cell proliferation into account marked by *.

**Figure 4 ijms-20-01306-f004:**
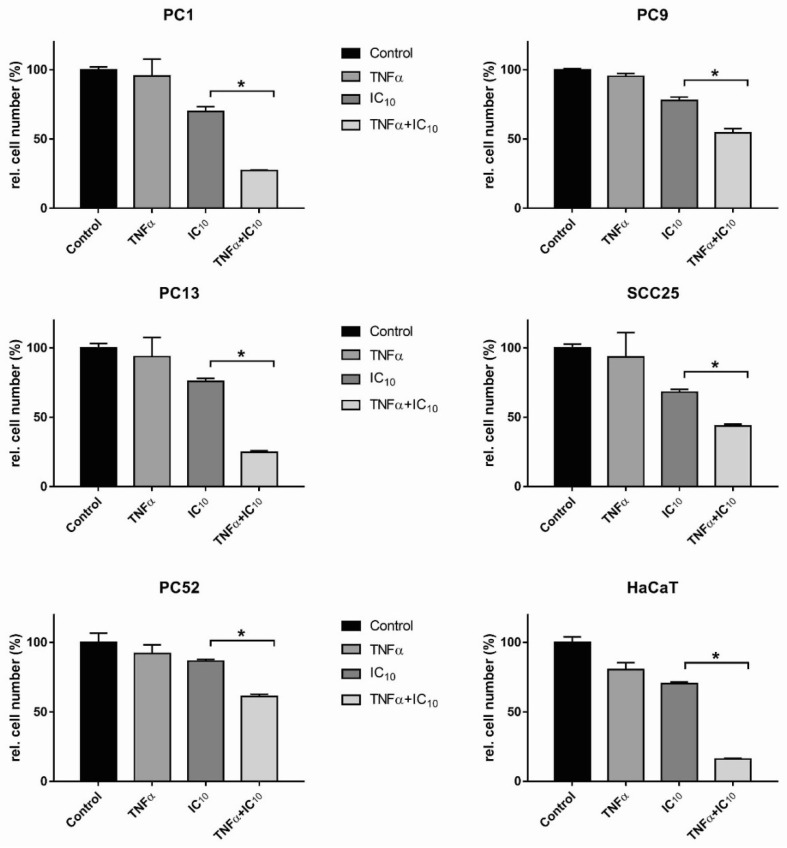
Relative cell number after stimulation with NFκB inhibitor TPCA1 in combination with TNFα. To determine whether the combination of TNFα and TPCA1 leads to a reduction in cell number via cell death, all cell lines (1 × 10^4^/well) were stimulated with TNFα (100 ng/mL) and the cell line-specific IC_10_ of TPCA1 for 72 h and stained with crystal violet. Data of one representative experiment are shown (*n* = 3). Results were analysed using the Wilcoxon rank-sum test. A significance level of *p* < 0.05 was established to indicate statistically significant effects and marked with *.

**Figure 5 ijms-20-01306-f005:**
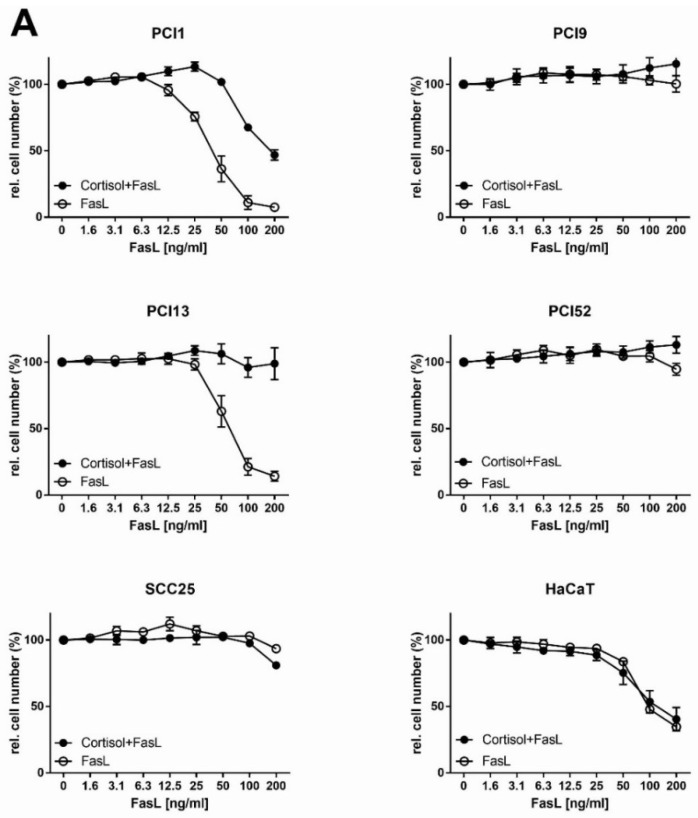
Combination of Cortisol and FasL in HNSCC cell lines. (**A**) HNSCC cell lines (1 × 10^4^/well) were incubated with the indicated concentrations of FasL in combination with the cell line-specific constant IC_10_ of Cortisol as indicated in Table 1. The RCN was analysed after 72 h via crystal violet staining, and values were normalised to that of the control (100%). These graphs show data of three independently reproduced experiments with the mean values and standard deviations (*n* = 3). Student’s *t*-test was used for the statistical evaluation of these results to show significant effects when comparing both therapies. To compare treatment efficacies, Tallarida’s interaction index was determined as indicated in the material and method section. (**B**) HaCaT cells (1 × 10^4^/well) were incubated with the indicated concentrations of FasL in combination with IC_10_ concentration of MLN4924, QNZ and TPCA1 as indicated in Table 1. The RCN was analysed after 72 h via crystal violet staining, and the values were normalised to that of the control (100%). Three independent experiments were carried out with triplicate cultures. These graphs show data of three independently reproduced experiments (*n* = 3) with the mean values and standard deviations.

**Table 1 ijms-20-01306-t001:** Cell line-specific IC_10_ and IC_50_ values for the NFκB inhibitors Cortisol, MLN4924, QNZ and TPCA1. Cells (1 × 10^4^/well) were stimulated for 72 h with the indicated concentrations of Cortisol, MLN4924, QNZ and TPCA1. The RCN was determined via crystal violet staining and normalised to that of the untreated control (100%). The IC_10_ and IC_50_ values were determined as described in the material and methods section. Three independent experiments were carried out to determine mean values (*n* = 3). Cell lines for which no IC_10_ or IC_50_ values could be determined are marked with *. IC_50_ values of inhibitors showing no effect or only a minimal effect are marked with **. In these cases, the maximum concentration is indicated. In PCI9 and PCI52, no IC_10_ could be determined for QNZ. For cell lines marked with **^†^**, 1 µM was defined as the IC_10_ value.

	Cortisol [µM]		MLN4924 [µM]		QNZ [µM]		TPCA1 [µM]	
	IC_10_	IC_50_	IC_10_	IC_50_	IC_10_	IC_50_	IC_10_	IC_50_
PCI1	20 *	>280 **	0.2	1	0.05	1	4.6	9.7
PCI9	140 *	>280 **	3.1 *	>4.8 *	1 ^†^	>10 **	3.3	13.8
PCI13	140 *	>280 **	0.2	0.8	0.4	1.2	5.2	24.2
PCI52	140 *	>280 **	7.3*	>17.4 *	1 ^†^	>10 **	6.3 *	>17 *
SCC25	140 *	>280 **	0.2	1.4	10 *	>10 **	3.5	6.1
HaCaT	140 *	>280 **	0.2	0.8	0.4	1.1	2	5.3

**Table 2 ijms-20-01306-t002:** Relative cell numbers (%) after 72 h of incubation with the highest concentrations of the NFκB inhibitors. The table shows the RCN value after incubation for 72 h with Cortisol (50 µg/mL), MLN4924 (100 µM), QNZ (10 µM) and TPCA1 (100 µM). The values agree with the values shown in Figure 2 at the highest indicated concentration. The ± columns show the standard deviation.

	PCI1		PCI9		PCI13		PCI52		SCC25		HaCaT	
	RCN	±	RCN	±	RCN	±	RCN	±	RCN	±	RCN	±
Cortisol	62	6.6	75.5	8.8	71.3	2.8	86.9	0.7	91	2.3	72.6	4.6
MLN4924	9.9	4.2	39.3	4.6	4.4	1.9	35.6	4.1	15.0	2.5	18.5	5.5
QNZ	5.7	3.9	96.3	3.2	6.0	5.7	77.0	5.6	81.2	3.6	39.1	8.2
TPCA1	11.5	2.3	29.9	5.2	18.0	6.3	36.2	2.1	10.6	3.9	18.5	8.4

**Table 3 ijms-20-01306-t003:** Efficiency quotient and interaction index (*y*) of combination therapy with NFκB inhibitors and the death ligand FasL. HNSCC cell lines (1 × 10^4^/well) were incubated with the indicated concentrations of FasL in combination with the cell line-specific constant IC_10_ of the indicated NFκB inhibitor (*n* = 3) as shown in Table 1. The values of the IC_50_ FasL_mono_ and IC_50_ FasL_combi_, efficiency quotient and interaction index were determined as described in the material and methods section. The interaction index, as described by Tallarida, for determining synergistic effects is annotated as follows: * = superadditive (synergistic, resp.), ** = additive, *** = antagonistic. Samples with pro-apoptotic sensitisation of the cells are highlighted in bold. Cell lines for which no IC_50_ could be determined due to FasL resistance (PCI9, PCI52 and SCC25) as well as cell lines that remained resistant (PCI9 and PCI52) are marked with ^†^. Furthermore, samples for which a complete IC_50_ saturation curve was not formed for FasL are designated with ^‡^. Samples for which the additional application of NFκB pathway inhibitors led to a decrease in the FasL effect are marked with ^±^.

	IC_50_ FasL [ng/mL]	Efficiency IC_50_ FasL_mono_/ IC_50_ FasL_kombi_)	Interaction Index (y)
**PCI1**			
FasL	37.3	-	-
Cortisol [IC_10_] + FasL	>88.8 ^±^	0.4 ^±^	2.4 ***
MLN4924 [IC_10_] + FasL	33.6	1.1	1.1 ***
QNZ [IC_10_] + FasL	44.3 ^±^	0.8 ^±^	1.2 ***
TPCA1 [IC_10_] + FasL	41.9 ^±^	0.9 ^±^	1.6 ***
**PCI9**			
FasL	- ^†^	-	-
Cortisol [IC10] + FasL	>200 ^†^	1	1.5 ***
MLN4924 [IC10] + FasL	>200 ^†^	1	1.7 ***
QNZ [IC10] + FasL	>200 ^†^	1	1.1 ***
TPCA1 [IC10] + FasL	>200 ^†^	1	1.2 ***
**PCI13**			
FasL	55.5	-	-
Cortisol [IC10] + FasL	>200 ^±^	0.3 ^±^	4.1 ***
MLN4924 [IC10] + FasL	>71.9 ^±^	0.8 ^±^	1.6 ***
QNZ [IC10] + FasL	**24.4**	**2.3**	**0.8** *
TPCA1 [IC10] + FasL	>55.1 ^±^	1	1.2 ***
**PCI52**			
FasL	- ^†^	-	-
Cortisol [IC_10_] + FasL	>200 ^†^	1	1.5 ***
MLN4924 [IC_10_] + FasL	>200 ^†^	1	1.4 ***
QNZ [IC_10_] + FasL	>200 ^†^	1	1.1 ***
TPCA1 [IC_10_] + FasL	>200 ^†^	1	1.4 ***
**SCC25**			
FasL	- ^†^	-	-
Cortisol [IC_10_] + FasL	>200 ^†^	1	1.5 ***
MLN4924 [IC_10_] + FasL	**39.7**	**5**	**0.3** *
QNZ [IC_10_] + FasL	>200 **^†^**	1	2 ***
TPCA1 [IC_10_] + FasL	>200 **^†^**	1	1.6 ***
**HaCaT**			
FasL	>68.7 **^‡^**	-	-
Cortisol [IC_10_] + FasL	>59.1	1.2	1.4 ***
MLN4924 [IC_10_] + FasL	**17.6**	**3.9**	**0.5** *
QNZ [IC_10_] + FasL	**28.7**	**2.4**	**0.8** *
TPCA1 [IC_10_] + FasL	**27.5**	**2.5**	**0.8** *

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
