# Peer review of "The Selection of NFκB Inhibitors to Block Inflammation and Induce Sensitisation to FasL-Induced Apoptosis in HNSCC Cell Lines Is Critical for Their Use as a Prospective Cancer Therapy"

_ijms, 2019, doi:10.3390/ijms20061306_

Reviewer 1 Report

The authors examined the influence of four NFκB inhibitors cortisol, MLN4924, QNZ and

TPCA1 on proliferation, inflammation and sensitization to apoptosis mediated by the death

ligand FasL in the HNSCC cell lines PCI1, PCI9, PCI13, PCI52 and SCC25 and in the human dermal keratinocyte cell line HaCaT. Among the four inhibitors used in this study, they observed that only TPCA1 truly qualifies for clinical use in HNSCC. Thus, they suggested that in vivo studies are needed to demonstrate the efficacy of TPCA1 in HNSCC.

Major point

They should explain Why TPCA1 is the most effective in HNSCC, and why only TPCA1 does not interfere with FasL mediated apoptosis.

They should explain Why TPCA1 is the most effective in HNSCC, and why only TPCA1 does not interfere with FasL mediated apoptosis.

Minor point

Misspelling check and proof reading are required.

Author Response

Dear Reviewer,

First of all, many thanks for the review of our paper.

We did not give any precise explanation why TPCA1 works better, because we do not have the experimental evidence or hind. We think, that every reason is very speculative. Nevertheless, we inserted the following paragraph in the discussion based on your comment.

„But why is TPCA1 so effective compared to the other NFκB inhibitors? Based on the current state of knowledge, the effect can’t be explained by a detailed molecular mechanism. However, TPCA1 attacks the IKK2 complex specifically, while the other inhibitors affect only indirectly the IκBα phosphorylation and/or the NFκB pathway. We assume that in the analysed HNSCC cells TPCA1 is the only inhibitor that efficiently inhibits the classical NFκB pathway - TPCA1 alone was able to inhibit IL-8 secretion. The other drugs may affect the classical or alternative pathway, but not specifically. In HNSCC cells they are more responsible for an imbalance in NFκB-regulated gene expression and a rather unspecific cellular response. Furthermore, the influence of other signal transduction pathways is also important. Cortisol, in its active form, is a transcription factor that can activate many gene products, the exact mechanism of QNZ is unknown, and MLN4924 also does not specifically interfere via the proteasomal degradation. Thus, the main advantage of TPCA1 is its selectivity, which also contributes to the fact that this drug has no effect on FasL-mediated apoptosis in HNSCC cell lines.”

Additionally, several people have read the paper again to minimize spelling mistakes.

Best regards,

Axel Seher

Reviewer 2 Report

The manuscript by Scheurer et al. shows that the four NFκB inhibitors - cortisol, MLN4924, QNZ and TPCA1 block inflammation and induce sensitization to FasL4 induced Apoptosis in HNSCC cell lines. The authors found that TPCA1 was qualified as an excellent therapeutic mediator in HNSCC by four positive qualities: 1) Proliferation was inhibited at low μM-range concentrations; 2) TNFα-induced IL-8 secretion was blocked; 3) HNSCC cells were sensitized to TNFα-induced cell death; and 4) FasL-mediated apoptosis was not disrupted. The experimental data are convincing, however minor improvements needed before accepting the manuscript for the publication.

Specific Comments:

1.     Statistics is poorly conducted, “n” details are not provided for most of the experimental data. Please give the detailed number of times experiments repeated independently, a number of cells/dishes analyzed.

2.     Authors should include statistics in all the figures with a detail of the test used in the figure legends.

Author Response

Dear Reviewer,

First of all, many thanks for the review of our paper.

We have included detailed information about the statistics in both the material and method section as well as in the figure legends.

Best regards,

Axel Seher